# Towards Multi-Grained Explainability for Graph Neural Networks

**Xiang Wang**[§†‡], **Ying-Xin Wu**[§], **An Zhang**[†], **Xiangnan He**[§]*, **Tat-Seng Chua**[†]

[‡]Sea-NExT Joint Lab
[†]National University of Singapore
[§]University of Science and Technology of China
xiangwang@u.nus.edu, wuyxin@mail.ustc.edu.cn, an_zhang@nus.edu.sg
xiangnanhe@gmail.com, dcscts@nus.edu.sg

## Abstract

When a graph neural network (GNN) made a prediction, one raises question about explainability: "Which fraction of the input graph is most influential to the model's decision?" Producing an answer requires understanding the model's inner workings in general and emphasizing the insights on the decision for the instance at hand. Nonetheless, most of current approaches focus only on one aspect: (1) local explainability, which explains each instance independently, thus hardly exhibits the class-wise patterns; and (2) global explainability, which systematizes the globally important patterns, but might be trivial in the local context. This dichotomy limits the flexibility and effectiveness of explainers greatly. A performant paradigm towards multi-grained explainability is until-now lacking and thus a focus of our work. In this work, we exploit the pre-training and fine-tuning idea to develop our explainer and generate multi-grained explanations. Specifically, the pre-training phase accounts for the contrastivity among different classes, so as to highlight the class-wise characteristics from a global view; afterwards, the fine-tuning phase adapts the explanations in the local context. Experiments on both synthetic and real-world datasets show the superiority of our explainer, in terms of AUC on explaining graph classification over the leading baselines. Our codes and datasets are available at https://github.com/Wuyxin/ReFine.

## 1 Introduction

While graph neural networks (GNNs) [1, 2] have achieved great success in a variety of applications, they usually come as black-box models. The general problem about GNN explainability [3] is to answer "What knowledge does the model use to arrive at the conclusions in general and the specific decision at hand?". Thoroughly answering this question requires the global understanding of the model's inner workings and the local insights on a specific instance. Take a GNN model for molecular property prediction as an example. The global understanding exhibits the knowledge encoded in the model, such as the distribution of the chemical groups; meanwhile, the local insight identifies certain chemical groups responsible for a given molecule's property. Such multi-grained explainability flexibly and reliably inspects the decision-making process of the GNN [4, 5], which is critical to the applications on safety, fairness, and privacy [6, 7].

In the field of GNN explainability [8], explainer models broadly attribute model prediction to the input graph, then sample a salient subgraph as the explanation for the model prediction. However, most of current explainers focus on either on local [9, 10, 6, 11, 12] or global explainability [13, 7], thereby suffer from inherent limitations correspondingly:

---

*Xiangnan He is the corresponding author.

35th Conference on Neural Information Processing Systems (NeurIPS 2021).

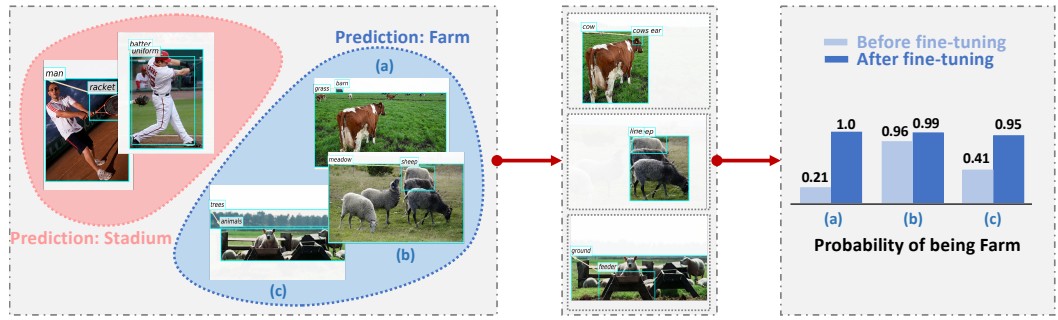

Figure 1: Explanations on Visual Genome dataset generated from ReFine, including the pre-training and fine-tuning phases. Right indicates the changes before and after the fine-tuning.

- Local explainability aims to customize the explanatory subgraph for each instance individually. However, such local explanations fall short in systematizing the prototypical patterns shared within a class or group of instances. Thus, they lack the global understanding of the model's workings [7, 13], which is vital to generalize to other instances being explained.

- Global explainability targets at the globally important patterns across multiple instances, which could violate the local fidelity [14] — the globally important substructure may not be important or even appear in the local context, thus might fail to explain a specific instance reliably.

Briefly put, these approaches overlook the multi-granularity nature of explainability, while we argue that the local and global explainability should be exhibited simultaneously to obtain faithful explanations. Taking Figure 1 as an example, the global explainability differentiates the explanations for various classes, such as *livestock-background* subgraphs for the farm class, *human-sports* subgraphs for the stadium class. When zooming in a specific scene graph, the local explainability refines on the farm-wise patterns and specifies *(sheep, on, meadow)* as the final explanation. A paradigm towards such multi-grained explainability is until-now lacking, to the best of our knowledge.

Towards multi-grained explainability, we propose a novel explainer, ReFine, with pre-training and fine-tuning [15, 16] techniques for explaining GNN models. Specifically, pre-training aims to answer "What class-wise knowledge does the GNN leverage to make predictions in general?". We combine the contrastive learning [17, 18] into class-wise generative probabilistic models [7], thereby approach coarser-grained explanations (*i.e.* saliency maps of all edges). Going beyond the global view, fine-tuning is to answer "Why the GNN model made the certain prediction for the instance at hand?", where we upgrade the coarser-grained explanations to the finer-grained explanations (*i.e.* explanatory subgraphs of salient edges). Through this way, ReFine can faithfully generate multi-grained explanations, and we empirically show its effectiveness as compared to some state-of-the-art explainers [9, 6, 7, 19]. It is also worth mentioning that, although the general understanding of GNN predictions has been considered in a recent work PGExplainer [7], it is only exploited to train a generative probabilistic model shared across all the explained instances, rather than dissecting and modeling the class-wise knowledge explicitly. Overall, our contributions are summarized as:

- We investigate the local explainability and global explainability for explaining GNNs and put forward the concept of multi-grained explainability.

- We propose a pre-training and fine-tuning framework to generate multi-grained explanations, which has both global understanding of model workings and local insights on specific instances.

- We achieve state-of-the-art performance on various datasets *w.r.t.* predictive accuracy on explaining GNNs. Quantitative and qualitative results verify multi-granularity explainability of ReFine.

## 2    Background & Task Formulation

In this section, we begin with the backgrounds on GNNs and frame the task of generating multi-grained explainability for GNN models.

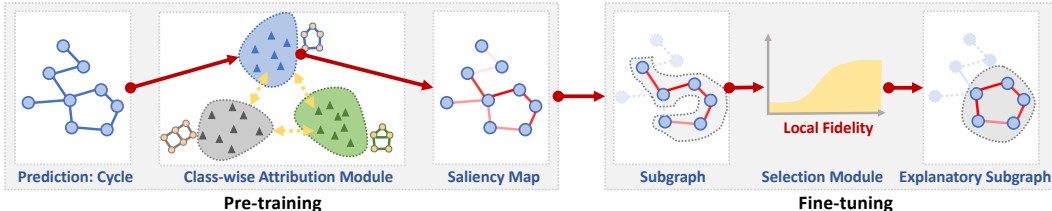

Figure 2: Model construction of proposed ReFine. Left represents the pre-training phase for a graph example, which is labeled and predicted as "Cycle", from the BA-3motif dataset. Right demonstrates the fine-tuning process where the saliency map is fine-tuned on the instance to achieve local fidelity.

**Graph Neural Networks.** We denote the graph data as $\mathcal{G} = (\mathcal{V}, \mathcal{E})$ with the node set $\mathcal{V}$ and the edge set $\mathcal{E}$. The structural feature of a graph can be represented by an adjacency matrix $\mathbf{A} \in \{0, 1\}^{|\mathcal{V}| \times |\mathcal{V}|}$, where $A_{ij} = 1$ indicates an edge starting from node $i$ to node $j$, and $A_{ij} = 0$ otherwise. The node feature matrix is represented as $\mathbf{X} \in \mathbb{R}^{|\mathcal{V}| \times d}$.

Graph neural networks (GNNs) [1, 2] aim to generate powerful representation on graphs in an end-to-end fashion. Such representation facilitates the downstream tasks, such as node classification [20, 21], link prediction [22, 23, 24, 25], and graph classification [26]. Without loss of generality, we consider a graph classifier $f : \mathbb{G} \to \mathbb{R}^C$, which classifies an input graph $\mathcal{G} \in \mathbb{G}$ in $C$ categories and outputs prediction by $c = \arg\max_i f(\mathcal{G})_i$. Typically, $f$ consists of three components: (1) learning of node representations, which distills vectorized information from neighboring nodes and updates node representations recursively; (2) learning of graph representation, which aggregates the node representations to establish the representation for the holistic graph; (3) graph classification, which maps the graph representation into the probability distribution of different categories.

**Explaining Graph Neural Networks.** The explainer model (*aka.* the explanation method) usually performs two consecutive operations: (1) feature attribution [27, 28], which associates each feature of an input $\mathcal{G} \in \mathbb{G}$ with the relevance score for the classifier's prediction; (2) feature selection [29, 6], which extracts salient features based on the relevance scores to construct an explanatory subgraph. The subgraph is regarded as the evidence for the GNN to make the prediction.

We follow previous works [6, 7, 10, 19] and focus on the contributions of the structural features (*i.e.* edges). Our explainer consists of two components: an attribution module $\mathcal{T}$ for edge attribution and a selection module $\mathcal{H}$ for edge selection. Specifically, $\mathcal{T}$ assigns the adjacency matrix $\mathbf{A}$ with a saliency map, *i.e.*

$$\mathbf{M} = \mathcal{T}(\mathcal{G}, f, c), \tag{1}$$

where $\mathbf{M} \in \mathbb{R}^{|\mathcal{V}| \times |\mathcal{V}|}$, each element of which is the importance score of the edge to the prediction class $c$. Such saliency map can further result in an attentive graph $\mathcal{G}_{att} = \mathbf{A} \odot \mathbf{M}$. Then, the selection module $\mathcal{H}$ identifies the edges of explanatory subgraph based on the attentive graph:

$$\mathbf{S} = \mathcal{H}(\mathcal{G}_{att}, f, c, \rho), \tag{2}$$

where $\mathbf{S} \in \mathbb{R}^{|\mathcal{V}| \times |\mathcal{V}|}$ constructs the explanatory subgraph $\mathcal{G}_{exp} = \mathbf{A} \odot \mathbf{S}$, and $\rho$ is the explanation budget [27] that equals to the number of nonzero elements in $\mathbf{S}$.

## 3 Methodology

Here we present our explainer that purses multi-grained explainability by pre-training and fine-tuning, as Figure 2 shows. In the pre-training phase, the attribution module distills the class-wise knowledge, which contrasts the salient structures based on the prediction, answering the question "Why did the GNN model assign a group of graphs with the same prediction?". In the next phase, the selection module goes beyond the class-wise knowledge and fine-tunes the saliency maps on a specific instance for answering "Why the GNN model made the certain prediction for the specific graph?".

### 3.1 Pre-training Towards Global Explainability

**Class-aware Attribution Module.** Towards the global explainability of GNN, it is important to specify the class-wise knowledge across the instances with the same prediction. Inspired by the success of generative models [7, 30, 31] in capturing the succinct structures from the graphs, we hire multiple generative probabilistic models [7] as our attribution models (short for attributor), *i.e.* $\mathcal{T}_\theta = \{\mathcal{T}^{(c)}|c = 1, \cdots, C\}$ which is parameterized by $\theta$. The attributor $\mathcal{T}^{(c)}$ is responsible for uncovering the hidden patterns from some graph instances $\mathcal{O}^{(c)} = \{\mathcal{G}|c = \arg\max_i f(\mathcal{G})_i\}$ with the same prediction class $c$.

Formally, each attributor $\mathcal{T}^{(c)}$ is composed of a GNN encoder GNN$^{(c)}$ and a MLP decoder MLP$^{(c)}$, whose parameters are shared when explaining graphs in $\mathcal{O}^{(c)}$, so as to systematize the class-wise patterns. Next we introduce the construction of each class-wise attributor, while we omit the superscript for conciseness. Specifically, the encoder GNN embeds each node $i$ in $\mathcal{G}$ with representation $\mathbf{z}_i$ and summarize the representations of all nodes as:

$$\mathbf{Z} = \text{GNN}(\mathcal{G}, \mathbf{X}), \tag{3}$$

where $\mathbf{Z} \in \mathbb{R}^{|\mathcal{V}| \times d'}$ encodes the structural feature $\mathbf{A}$ and node feature $\mathbf{X}$. On the top of the node representations, we model the graph structure as edge distributions and frame the generation of explanatory subgraphs by sampling from the edge distributions:

$$P(\mathbf{M}|\mathbf{Z}) = \prod_{(i,j)\in\mathcal{E}} P(M_{ij}|\mathbf{z}_i, \mathbf{z}_j), \tag{4}$$

where $M_{ij}$ indicates the importance of edge $(i, j)$. Then the MLP encoder takes the concatenation of node representations $\mathbf{z}_i$ and $\mathbf{z}_j$ as the inputs and outputs the importance score. To approximate the importance score to the discrete distribution and optimize the generator via gradient propagation, we adopt the reparameterization trick [7], where an independent random variable $\epsilon \sim \text{Uniform}(0, 1)$ is introduced. As such, the edge probability is formulated as:

$$P(M_{ij}|\mathbf{z}_i, \mathbf{z}_j) = \sigma((\log\frac{\epsilon}{1-\epsilon} + \alpha_{ij})/\beta), \quad \text{with} \quad \alpha_{ij} = \text{MLP}([\mathbf{z}_i, \mathbf{z}_j]), \tag{5}$$

where $\sigma$ is the sigmoid function, and $\beta$ denotes the temperature hyperparameter. It is worth emphasizing that our attributors is different from PGExplainer [7], where only one generative probabilistic model is involved. Thus, their attribution results are limited in differentiating the patterns of different classes and systematizing the class-wise knowledge.

**Pre-training Class-wise Attribution Module.** We devise the following objective function for training the class-wise attributors.

$$\min_\theta \mathcal{L}_1 + \gamma\mathcal{L}_{cts}, \tag{6}$$

where $\gamma$ is the trade-off hyperparameter. We start from maximizing the mutual information between the attentive graphs and the target prediction of the graph, which is a widely-used learning paradigm in the literature [32, 6, 7]. It guides us to find the prediction-relevant explanatory subgraph, which equals to minimizing the following loss:

$$\mathcal{L}_1 = -\mathbb{E}_{\mathcal{G}}\mathbb{E}_{\epsilon}\mathbb{E}_{c'}[P(Y = c'|G = \mathcal{G})\log P(Y = c'|G = \mathcal{G}_{att}^{(c)})], \tag{7}$$

where $G$ and $Y$ are the graph and prediction variables, respectively; $\mathcal{G}$ is the full graph instance to explain; by sampling $\epsilon \in \text{Uniform}(0, 1)$ and $c' \in \{1, \cdots, C\}$, the class-wise saliency map $\mathbf{M}^{(c)}$ can be generated from Equation (4); $P(Y = c'|G = \mathcal{G}) = f(\mathcal{G})_{c'}$ is the output probabilities of the prediction being $c'$ when feeding $\mathcal{G}$ to the GNN model $f$; analogously, $P(Y = c'|G = \mathcal{G}_{att}^{(c)}) = f(\mathcal{G}_{att}^{(c)})_{c'}$ audits the output probability when feeding $\mathcal{G}_{att}^{(c)} = \mathbf{A} \odot \mathbf{M}^{(c)}$.

Moreover, we introduce a contrastive learning [33, 34, 18, 35, 36, 37] loss to emphasize differences among the class-wise patterns — the substructure of the full graph that is distant to that of the graphs with a different prediction but close to that of the graphs with the same prediction. It makes each attributor focus on the unique and discriminative information within the class. Specifically, for the

saliency maps $\mathcal{G}_{att1}^{(c_1)}$ of $\mathcal{G}_1$ and $\mathcal{G}_{att2}^{(c_2)}$ of $\mathcal{G}_2$, it encourages the agreements between $\mathcal{G}_{att1}^{(c_1)}$ and $\mathcal{G}_{att2}^{(c_2)}$ when $c_1 = c_2$, compared to that when $c_1 \neq c_2$:

$$\mathcal{L}_{cts} = \mathbb{E}_{\mathcal{G},\mathcal{G}'}\mathbb{E}_{\epsilon,\epsilon'}[(-1)^{\mathbb{I}(c_1=c_2)} \times \mu(\ell(\mathcal{G}_{att1}^{(c_1)}, \mathcal{G}_{att2}^{(c_2)}))], \tag{8}$$

where $\mu$ is the softplus function [34]; $\ell$ measures the similarity between two subgraphs, which is set as the representation similarity — $\ell(\mathcal{G}_{att1}^{(c_1)}, \mathcal{G}_{att2}^{(c_2)}) = \mathbf{h}_1^\top \mathbf{h}_2$ where $\mathbf{h}_1$ is the graph representations by feeding $\mathcal{G}_{att1}^{(c_1)}$ into the encoder $\text{GNN}^{(c_1)}$ and aggregating the node representations. Similar for $\mathbf{h}_2$. In addition, following [6], we adopt the element-wise entropy and $L_1$ norm on the edge probability. By jointly optimizing these two losses in Equation (6), the class-wise attribution module learns to stratify the discriminative information for different classes and generate the saliency maps with a global view of the target GNN. Taking an information-theoretical look at Equation (8), minimizing contrastive learning loss is maximizing a lower bound of the mutual information between the latent graph representations of two graphs within the same class.

## 3.2  Fine-tuning Towards Local Explainability

Having established the saliency map that exhibits the importance of each edge, the standard way is to rank all edges based on their importance scores and simply select the top edges as the explanatory subgraphs. However, we argue that such a coarser-grained selection fails to consider the dependencies of these selected edges explicitly. Within a high-quality explanatory subgraph, edges are supposed to cooperate with each other, form the coalition, and approach the target prediction better than individuals [38, 39]. Without considering such coalition effect, the quality of the explanatory subgraph is greatly limited.For example, when explaining why the molecule graph is classified as mutagenic [13], two connected nitrogen-oxygen (N-O) bonds form a chemical group $NO_2$ and present more discriminative information about the mutagenic property [13]; whereas, two salient but disconnected N-O bonds from different chemical groups are less informative to interpret the mutagenic property.

Clearly, the coarser-grained saliency maps are insufficient to exhibit the coalition effect of edges, thus might be redundant and suboptimal explanations. Hence, we move forward to learn a finer-grained explanatory subgraph. Technically, on the top of the well-trained class-wise attribution module, we add the selection module:

$$\mathbf{S}^{(c)} = \mathcal{H}(\mathcal{G}_{att}^{(c)}, f, c, \rho), \tag{9}$$

where $\rho$ is the number of edges selected in the explanatory subgraph; $\mathcal{H}$ is a sampling (selection) function; $\mathbf{S}^{(c)}$ preserves the elements selected by the selection function and sets the other elements as $0$. Instead of the hard selection that picks up the edges with the highest probability, $\mathcal{H}$ samples edges according to their probabilities. Allowing edges with low probabilities to be sampled can prevent the explainer from collapsing to suboptimal solutions with limited coalition effect.

With the new stochastic adjacency matrix $\mathbf{S}^{(c)}$, we are able to extract the subgraph $\mathcal{G}_{exp}^{(c)}$. To fine-tune the attribution and selection modules, we resort to maximize the mutual information between the explanation candidate $\mathcal{G}_{exp}^{(c)}$ and the target prediction of the full graph:

$$\mathcal{L}_2 = -\mathbb{E}_{\mathcal{G}}\mathbb{E}_{\epsilon}\mathbb{E}_{c'}[P(Y = c'|G = \mathcal{G}) \log P(Y = c'|G = \mathcal{G}_{exp}^{(c)})]. \tag{10}$$

By optimizing the loss above, the selection module accounts for the edge coalition within $\mathbf{S}^{(c)}$, so as to achieve higher local fidelity. Moreover, as the selection module discards some elements in the stochastic adjacency matrix, it blocks parts of gradient backpropagation and possibly acts as a dropout function to avoid the overfitting on the instance-level explanations.

## 4  Experiments

We mainly aim to investigate the following questions:

- **RQ1**: How effective is the pre-training phase of ReFine, as compared to that of existing methods?

- **RQ2**: How effective is the fine-tuning phase of ReFine, as compared to that of the pre-training phase?

### 4.1 Experimental Settings

**Datasets and Target GNNs**. We consider four datasets with various target GNNs:

- **Molecule graph classification.** We use the Mutagenicity dataset [40, 41], where $4,337$ molecule graphs are classified into two classes based on their mutagenic effect on a bacterium. The well-trained Graph Isomorphism Network (GIN) [26, 42] has achieved a 100% testing accuracy.
- **Scene graph classification.** Following the previous work [10], we select $4,443$ (images, scene graphs) pairs from Visual Genome [43] to construct the VG-5 dataset. Wherein, the graphs are labeled with five classes: stadium, street, farm, surfing, forest. Each graph contains regions of the objects as the nodes, while edges indicates the relationships between object nodes. The target GNN is an APPNP [44] which achieves 64.3% testing accuracy.
- **Handwriting graph classification.** We use the MNIST superpixel dataset [45], which converts 70,000 images into the graphs of superpixel adjacency. Every graph is labeled as one of ten digit classes. We trained a Spline-based GNN [46] which gains 97.9% accuracy in the testing dataset.
- **Motif graph classification.** We follow prior studies [6, 7] to create a synthetic dataset, BA-3motif, which contains 3,000 graphs. Specifically, we adopt the Barabasi-Albert (BA) graphs as the base, and attach each base with one of three motifs: house, cycle, grid. The trained GNN model, ASAP [47], classifies them according to the type of attached motifs and achieved 100% testing accuracy.

**Baselines.** We compare our ReFine with the state-of-the-art explanation methods:

- **SA** [9] directly uses the gradients of the model prediction *w.r.t.* the adjacency matrix of the input graph as the importance of edges.
- **GNNExplainer** [6] applies the soft masks on the messages carried by edges, where each mask indicates an edge's importance. Note that the masks of graph instances are trained individually.
- **PGExplainer** [7] hires a neural network to learn to generate the masks for the input edges. The generative model is trained over multiple explained instances.
- **PGM-Explainer** [19] collects the prediction change on the random node perturbations, and then learns a Bayesian network from these perturbation-prediction observations, so as to capture the dependencies among the nodes and the prediction. Here we transfer it to model the edge importance.

**Optimization.** For the parametric explanation methods (GNNExplainer, PGExplainer, PGM-Explainer), we apply a grid search to tune their own hyperparameters. For our ReFine framework, we use the Adam optimizer and set the learning rate of pre-training and fine-tuning as 1e-3 and 1e-4, respectively. All experiments are done on a single Tesla V100 SXM2 GPU (32 GB).

**Evaluation Metrics.** It is challenging to quantitatively evaluate the quality of explanations, since the ground-truth explanations are usually unavailable. In the literature, there are three widely-used evaluation metrics:

- **Predictive Accuracy (ACC@$\rho$)** [32, 48, 27]. It measures the fidelity of the explanatory subgraphs by feeding it solely into the target model and auditing how well it recovers the target prediction. We report the average ACC@$\rho$ over all graphs in the testing sets, and further denote ACC-AUC as the area under the ACC curve over different selection ratios $\rho \in \{0.1, 0.2, \cdots, 0.9, 1.0\}$. ACC@$\rho$ and ACC-AUC are suitable for all the datasets.
- **Recall@$N$.** As suggested in prior studies [6, 7, 32], we can create the "ground-truth explanations" for the synthetic dataset. Specifically, for BA-3motif, the motif of each graph can be viewed as the discriminative information coherent in the model knowledge. As such, we can frame the evaluation problem as the task of top edge ranking. To be more specific, for an explanatory subgraph, the edges within the motif are positive, while the others are negative. To this end, recall can be adopted as the evaluation protocols. More formally, Recall@$N = \mathbb{E}_{\mathcal{G}}[|\mathcal{G}_s \cap \mathcal{G}_s^*|/|\mathcal{G}_s^*|]$ where $\mathcal{G}_s$ is composed of the top-$N$ edges and $\mathcal{G}_s^*$ is the ground-truth explanatory subgraph.

### 4.2 Quantitative Evaluations

**Influence of Pre-training (RQ1).** To investigate the effectiveness of pre-training, we first compare the performance of the attribution module with the state-of-the-art explainers. We denote this variant

Table 1: Structure/Training Difference of PGExplainer, ReFine and its ablation models.

| | Pre-training | | Fine-tuning |
| --- | --- | --- | --- |
| | Class-wise Attributors | Contrastive Learning | |
| PG-Explainer | - | - | - |
| Refine-CT | ✔ | - | - |
| Refine-FT | ✔ | ✔ | - |
| Refine | ✔ | ✔ | ✔ |

Table 2: Comparison of our ReFine and other baseline explainers

| | Mutagenicity | VG-5 | MNIST | BA-3motif | |
| --- | --- | --- | --- | --- | --- |
| | ACC-AUC | ACC-AUC | ACC-AUC | ACC-AUC | Recall@5 |
| SA | 0.769 | 0.769 | 0.559 | 0.518 | 0.243 |
| GNNExplainer | 0.895±0.010 | 0.895±0.003 | 0.535±0.013 | 0.528±0.005 | 0.157±0.002 |
| PG-Explainer | 0.631±0.008 | 0.790±0.004 | 0.504±0.010 | 0.586±0.004 | 0.293±0.001 |
| PGM-Explainer | 0.714±0.007 | 0.792±0.001 | 0.615±0.003 | 0.575±0.002 | 0.250±0.000 |
| **ReFine**-CT | 0.888±0.008 | 0.891±0.002 | 0.526±0.007 | 0.610±0.004 | 0.248±0.001 |
| **ReFine**-FT | 0.945±0.011 | 0.906±0.002 | 0.587±0.008 | 0.616±0.003 | 0.299±0.002 |
| **ReFine** | **0.955**±0.005 | **0.914**±0.001 | **0.636**±0.003 | **0.630**±0.006 | **0.304**±0.000 |
| Relative Impro. | 6.7% | 2.1% | 3.4% | 7.5% | 3.8% |

by ReFine-FT, which disables the fine-tuning phase and simply constructs the explanatory subgraphs based on the saliency scores. Moreover, we build another variant ReFine-CT, which removes the contrastive loss (Equation (8)) from the pre-training phase, to study the effect of the contrastive loss on the class-wise knowledge modeling. To be more clear, we present the difference of PGExplainer [7], ReFine and its ablation models in Table 4.2. Table 2 presents the performance comparisons, from which we have several findings:

- ReFine-FT outperforms the baseline explainers in most cases. To be more specific, it achieves significant relative improvements over the strongest baselines *w.r.t.* ACC-AUC by 5.6% and 5.1% in Mutagenicity and BA-3motif, respectively. This demonstrates the rationality and effectiveness of the attribution module. We attribute these improvements to the class-wise knowledge modeling: (1) By specifying the attributor models for each class, ReFine-FT is able to capture the underlying patterns shared across the instances within the same class; and (2) Conducting the contrastive learning between different class-aware attributors makes ReFine-FT better stratify the discriminative information for different classes. The class-wise knowledge endows ReFine-FT with the global view of the target model's workings.

- Although PGExplainer is also equipped with the global view of the target model, its performance is worse than that of ReFine-FT. We ascribe this to the limitations of PGExplainer's global view, which is founded upon all the explained instances, but fails to differentiate the class-wise patterns. This again verifies the rationality and effectiveness of our attribution module.

- ReFine-FT outperforms ReFine-CT by a large margin, indicating that the contrastive learning plays a critical role in exhibiting the class-wise knowledge. Specifically, it summarizes the patterns across similar instances and focuses on the information pertinent to specific classes, while filtering the irrelevant and redundant information out.

- Interestingly, in MNIST, the result of ReFine-FT is worse than that of PGM-Explainer. One possible reason is that the random perturbations in PGM-Explainer create a collection of broken graphs and offer a more comprehensive observation of the graphs. We leave the exploration of subgraph-prediction relations as future work.

**Influence of Fine-tuning (RQ2).** To justify the effectiveness of the fine-tuning phase, we report the performance of ReFine with our selection module in Tables 2 and 3, as compared to the performance before fine-tuning. We have the following observations:

Table 3: Performance under different selection ratios before and after fine-tuning.

| ACC@$\rho$ | Mutagenicity | | VG-5 | | MNIST | | BA-3motif | |
|---|---|---|---|---|---|---|---|---|
| | 0.4 | 0.6 | 0.4 | 0.6 | 0.4 | 0.6 | 0.4 | 0.6 |
| ReFine-FT | 96.8% | 94.0% | 91.3% | 91.4% | 41.4% | 61.4% | 36.0% | 65.7% |
| ReFine | 97.8% | 96.2% | 92.2% | 93.4% | 71.4% | 82.0% | 39.0% | 72.8% |
| Improvement | +1.0% | +2.2% | +0.9% | +2.0% | +30.0% | +20.6% | +3.0% | +7.1% |

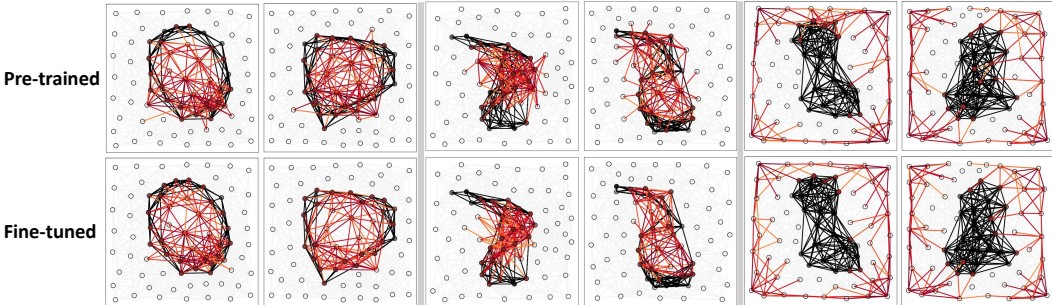

Figure 3: Qualitative Results in MNIST Superpixels dataset. Handwriting graphs are in black, which respectively represent number "0", "2", "8" within each block from left to right. Explanatory graphs are in red, where the top 10% edges are highlighted.

- Fine-tuning with the selection module can improves the explanation performance sustainably, which indicates the effectiveness of our pre-training and fine-tuning paradigm. Specifically, in MNIST, the predictive accuracy of the explanations after fine-tuning improves from $41.4\%$ to $71.4\%$ when the selection rato is $0.4$. We attribute these improvements to the local insights on specific instances: (1) Benefiting from the saliency map obtained in the pre-training phase, the selection module is able to filter noisy edges out and narrow down to where the target model looks to make decisions; (2) Fine-tuning the explanatory subgraphs considers the coalition effect of edges, thus approaches more information to recover the target prediction.

- Jointly analyzing Tables 2 and 3, ReFine consistently outperforms all baselines across the four datasets. Advantageous to the local or global explanations, our multi-grained explanations not only have the global understanding of model workings (*i.e.* the class-wise knowledge), but also account for the local insights on specific instances (*i.e.* the coalition effect of edges in the local context). It illustrates the superiority of our ReFine paradigm.

Overall, the empirical supports justify the significance of fine-tuning well. The contributions of fine-tuning *w.r.t.* the overall improvements over PG-Explainer are 37.1% and 31.8% in MNIST and BA-3motif datasets, respectively. One possible reason that fine-tuning contributes only 3.1% and 6.4% portion of overall improvements in Mutagenicity and VG-5 as compared to PG-Explainer is the existance of rich node features in these two datasets. With the assistance of node features, the global patterns might be well-captured during pre-training, thus leaving little space for the local patterns to improve.

### 4.3 Qualitative Analysis

We present the qualitative results on MNIST superpixel in Figure 3, where the pre-trained and fine-tuned explanations are the explanatory subgraphs before fine-tuning (*i.e.* extracted based on the saliency map) and after fine-tuning (*i.e.* derived from the selection module), respectively.

**Influence of Pre-training (RQ1).** The pre-trained results (first row) well demonstrate the global patterns, where the explanatory subgraphs for interpreting the digit "0" focus more on the edges between hollows in the middle and the fringe of the number. While interpreting the prediction "5", the explanations identify the edges spread on the bend of the number as the most important features. Also, we observe an interesting pattern in the results for explaining the prediction "8", where the

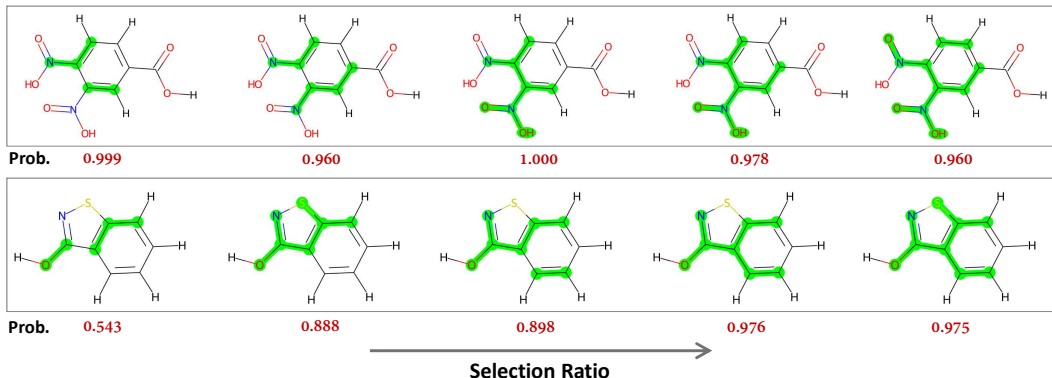

**Selection Ratio**

Figure 4: Qualitative Results in Mutagenicity dataset. The prediction of the molecule in the first row is mutagenic, while the molecule in the second row is predicted as non-mutagenic. The selection ratios range from 10% to 50%. Note that some opposite edges are visually coincident.

Table 4: Time costs (in second) of GNNExplainer, PG-Explainer and the fine-tuning phase of Refine.

|                    | Mutagenicity | VG-5  | MNIST | BA-3motif |
|--------------------|--------------|-------|-------|-----------|
| GNNExplainer       | 2.03         | 1.88  | 0.637 | 1.11      |
| PG-Explainer       | 0.030        | 0.035 | 0.040 | 0.032     |
| Refine(Fine-tuning)| 0.821        | 0.583 | 0.535 | 0.423     |

background edges draw more attention, rather than edges relevant to the digits, revealing the evidence for the target GNN to classify. It also shows the supporting evidence of the difference between the model explanation and the human explanation which focuses more on the digit graphs other than the background graphs. Through the pre-trained examples, the global patterns offer vital model understanding and inspections for the model's decision-making process.

**Influence of Fine-tuning (RQ2).** We now compare the pre-trained and fine-tuned explanations. Clearly, the fine-tuned explanatory graphs make clearer boundaries on the instances. The explanation adapted with the user-defined ratio pays greater attention to details that are only applicable to the specific instances. For example, one can take a closer look at the explanations in the 4-th column. Without the fine-tuning phase, the explanation may distracted by the edges across the digit and the background, such that these transition edges might be deemed as the most important features while achieve suboptimal predictive accuracies. In contrast, the fine-tuned explanation dispels such misunderstanding, with a higher local accuracy. Similar patterns can be found in other examples.

The qualitative results on Mutagenicity are presented in Figure 4, where each explanation has been fine-tuned on the corresponding ratio. We can see the flexibility on ReFine, which enables the fine-tuning on a specific user-defined ratio. With the selection ratio increases, the class probability output by the target GNN is generally stable or further improved. Moreover, the fine-tuning phase focuses more on the combination of features, with the constraint of selection ratio, to purse the higher accuracy rather than intercepting on a ranking based on the static edge importance, which is only valid under the addictive feature assumption [32].

## 4.4 Discussions

**Efficiency for Generating Explanations.** The inference time [7] to explain a new instance by the pre-trained ReFine is the same as PGExplainer under the same attributor construction. Different from GNNExplainer which has to retrain the model for each graph, ReFine only needs a few fine-tuning steps on the pre-trained model (20 steps on average). Thus, ReFine can gain a boosting performance for explaining graphs while remaining efficient in terms of time complexity. Specifically, we summarize the time costs in the Table 4. Clearly, our ReFine is more efficient than GNNExplainer and is computationally comparable to PG-Explainer.

**Limitations.** Although ReFine can well-encode the class-wise knowledge by learning the parameters of multiple attributors, it can hardly map such knowledge to the structure representation as XGNN [13]. This limits the human understanding on the core of input data via a conciseness substructure.

## 5 Related Work

We consider two classes of related work for GNNs explainability: studies on local explainability, which independently explain for each input graph without referring to other knowledge, *e.g.,* training data; studies on global explainability, which provide explanations for multiple instances with the guide of the model-level or class-level knowledge. See [49, 8, 50] for more overviews.

- **Local Explainability.** In general, there are two research lines. (1) Non-parametric explanation methods [10, 9, 11] use some heuristics as the feature contributions of a specific instance, without involving additional trainable models. Gradient-like scores [10, 9, 11] are wisely-used heuristics, which is obtained by backpropagating the model prediction or loss to the input features, such as adjacency matrix [10], along with the model architecture. (2) Parametric explanation methods [6, 19, 51, 52] additionally train a parametrized explainer model to generate the saliency maps or explanatory subgraphs for individual instances. The explainer model is typically optimized towards local fidelity [32, 48, 27], which uses the explanations to recover the target predictions. For example, GNNExplainer [6] learns soft masks for an instance and applies them on the adjacency matrix. PGM-Explainer [19] trains an Bayesian network upon the pairs of graph perturbations and prediction changes. However, these methods fall short in capturing the prototypical patterns shared within the same groups or classes.
- **Global Explainability.** This direction is less explored compared to the local explainability of GNNs [8]. To provide a global understanding of the model prediction, PGExplainer [7] formulates the generation of multiple explanations based on its collective and inductive property, and designs the attributor as a deep neural network whose parameters are shared across the explained instances. XGNN [13] explains GNNs by training a graph generator, which outputs class-wise graph patterns to explain this class. As it is designed to explain the holistic class, making it hardly applicable on an specific instance, *e.g.,* the graph patterns may not even exit on the instance.

## 6 Conclusion and Future Work

Multi-grained explainability promises to offer a flexible and all-round inspection of deep models' decision-making, which has been less explored in the literature. Motivated by this, we proposed a novel generative probabilistic model, ReFine, to approach the multi-granularity explainability via pre-training and fine-tuning. To exhibit global explanations with the prototypical patterns, the pre-training phase is founded upon the class-aware attribution modules and distills the class-level knowledge by contrastive learning. When given a specific instance, the fine-tuning phase further adapts the global explanations in the local context with high fidelity. In the fashion of pre-training and fine-tuning, we can generate explanations with both global patterns and local features. Extensive results in four datasets show that our method indeed improves the quality of explanatory subgraphs.

As future direction, we consider the extension of ReFine to fulfill the counterfactual explanation [53], which answers 'Why the target GNN model made a certain prediction, rather than another prediction?', to enrich the multi-granularity explainability. Further, multi-grained explainability can be exhibited to explore the model robustness and heuristically guide the model construction.

## Acknowledgments and Disclosure of Funding

Funding in direct support of this work: the Sea-NExT Joint Lab, Singapore MOE AcRF T2; the National Natural Science Foundation of China (U19A2079, 62121002); the National Key Research and Development Program of China (2020YFB1406703).

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
