# OpenReview forum: "Towards Multi-Grained Explainability for Graph Neural Networks"
_NeurIPS.cc/2021/Conference — NeurIPS 2021 Poster_

### Official Review · Reviewer_AmNU · 2021-07-14

**Rating:** 6
**Confidence:** 4

**Summary:**

This work proposes an instance-level explanation method called ReFine for graph neural networks. ReFine first generates a probability map over the edges in a similar way as PGExplainer. Then, it uses contrastive loss to encourage the graph embedding of explanations to be class distinctive. Next, the usage of the selection module makes the soft selection of the edges in the explanations.

**Ethics Review Area:**

["I don’t know"]

**Limitations And Societal Impact:**

YES

**Main Review:**

This work has a clear logic flow and strong motivations for each component in this method.

The usage of contrastive loss is creative and full of insight. And the ablation study in Table 1 also shows the effectiveness of this loss.

For the experiments, it provides solid results on both synthetic datasets and real-world datasets with different graph neural networks.

Enhancement: This work only provides the experiments on the graph classification task, while I believe it can also be applied to the node classification task. However, it would be better to also conduct experiments for node classification.

The major concern is the technical contribution. Compared with PGExplainer, the only difference is the proposed loss functions. Then the technical contribution is somehow limited.

I have a question about this work.

1. Through the experiment in Table 1, finetuning procedure shows its power. But I am still confused about the details of the fine-tuning procedure described in Section 3.2. Could you provide more details and insights about this fine-tuning procedure?

Overall, I think the idea is interesting and this paper is solid while the novelty is somehow limited. Hence, I would recommend a weak acceptance.


**Time Spent Reviewing:**

2

---

> ### Author Response · Authors · 2021-08-10
> **Response to Reviewer AmNU**
>
> Thank you for your comments. We're very glad you rate our work as well-motivated and creative. Please find our responses to your comments.
>
> **[Comment 1. Experiments for Node Classification]**
>
> Thanks for the suggestions on experiments for node classification. As our ReFine framework is task-agnostic, it can be applied to other graph learning tasks, such as node classification, graph matching. We will explore our framework on these tasks in future work.
>
> **[Comment 2. Technical Contribution & Difference from PG-Explainer]**
>
> We respectfully disagree with your statement that "compared with PGExplainer, the only difference is the proposed loss functions". Below, we clarify the difference between our ReFine and PG-Explainer.
>
>
> In the table below, we summarize the technical differences among PG-Explainer and ReFine family. Specifically, PG-Explainer only uses one universal attributor for all graph instances without distinguishing instances from different classes. Compared with PG-Explainer, ReFine has three major differences: (1) multiple class-wise attributors, each of which aims to capture the class-wise knowledge from the graph instances within the same class, rather than the whole population of graphs; (2) contrastive learning, which endows the class-wise knowledge with better discrimination ability; and (3) fine-tuning, which adaptively refine the class-wise knowledge towards better local fidelity for a single graph instance. Hence, our ReFine framework significantly differs from PG-Explainer.
>
> | | class-wise attributors | contrastive learning | fine-tuning                   |
> |---|------|-----|----------|
> | PG-Explainer | - | - | - |
> | Refine-CT | $\surd$ | - | - |
> | Refine-FT |$\surd$|$\surd$|-|
> | Refine |$\surd$|$\surd$|$\surd$|
>
> We further offer empirical evidence of each component's importance. Here we extract the information from Table 1 in our paper and calculate the contribution scores to ReFine's improvements over PG-Explainer in the table below. Obviously, the three components (class-wise attributors, contrastive learning, and fine-tuning) collaborate well, and all offer significant contributions.
>
>
> | | Mutagenicity | VG-5    | MNIST   | BA-3motif |   Average |
> |---|-----|---------|---------|-----------|---------|
> | contribution of class-wise attributors | 79.3\%       | 81.5\%  | 16.7\%  | 54.5\%    | 58.0\%  |
> | contribution of contrastive learning  | 17.6\%       | 12.1\%  | 46.2\%  | 13.6\%    | 22.4\%  |
> |contribution of fine-tuning | 3.1\%        | 6.4\%   | 37.1\%  | 31.8\%    | 19.6\%  |
>
>
> **[Comment 3. Fine-tuning Procedure]**
>
> We will offer a clearer description of the fine-tuning procedure. In Equation (9), we formulate the the selection module as $\mathbf{S}^{(c)}=\mathcal{H}(\mathcal{G}_{att}^{(c)}, f, c, \rho, t)$, where $\rho$ is the user-defined edge number. Specifically, when computing ACC-AUC, we set $\rho$ equal to 10\%,$\ldots$,100\% of the total number of edges, respectively.
>
> The sampling function $\mathcal{H}$ samples edges from a probability distribution, which is normalized from the saliency map of the attentive graph $\mathcal{G}_{att}^{(c)}$. We follow the simulated annealing algorithm, where the probability for $\mathcal{H}$ to accept a "worse" edge is proportional to $e^{-at}$, where $a$ is a hyper-parameter.

---

> > ### Comment · Reviewer_AmNU · 2021-08-23
> > **Thanks for the response.**
> >
> > I have read the author's response and my concerns are mostly addressed. I am keeping my score unchanged.

---

### Official Review · Reviewer_gLGL · 2021-07-14

**Rating:** 7
**Confidence:** 3

**Summary:**


This paper proposed a pretrain-fine tune framework for multi-grained GNN explaination. The experimental results show the proposed method to be effective.


**Limitations And Societal Impact:**

Yes.

**Main Review:**

Advantages:
1.	Experiments on multiple datasets are conducted to evaluate the method
2.	The proposed method is reasonable and easy-to-understand.
3.	The method is practical and easy for implementation.

Disadvantages:
1.	The proposed method is a combination of existing techniques. Specifically, the pre-training phase seems to be adopting PGExplainers for each class and adding a loss function to constrain the difference between classes. The fine-tuning phase is adopting simulated annealing algorithm to select subgraphs and use the subgraph to further train the model. This limits the technical contribution of this work.
2.	The authors give no theoretical analysis to their model. As the authors only made limited adjustments to existing works, even though the method is effective, more theoretical insights should be shown to provide understanding on how each part improves the performance and give inspirations to the research community.
3.	There are typos in the paper, such as ``Figure 3.1’’ in line 98.


**Time Spent Reviewing:**

8

---

> ### Author Response · Authors · 2021-08-10
> **Response to Reviewer gLGL**
>
> We sincerely appreciate your comments. Please find our responses to your comments and we hope they can resolve all your concerns.
>
> **[Comment 1. Novelty]**
>
> Here we clarify more about our novelty from several perspectives:
>
> 1. **Task Novelty**. Distinct from previous efforts either on global explainability or local explainability, our work focuses on multi-grained (multi-level) explainability that is less explored in the literature.
> 2. **Paradigm Novelty**. Towards multi-grained explainability, we introduce the pre-training & fine-tuning paradigm in the explanation generation, which seamlessly unifies the global and local explainability. Therefore, our main technical contribution lies in the novelty of tasks and paradigms.
> 3. **Novelty in Explainability**. There exist a handful of solutions towards class-aware knowledge. In our pre-training phase, we adopt contrastive learning, the simple but powerful tool, to achieve this goal. More importantly, although contrastive learning is used in graph learning, it has never been used in the literature of explainability, to the best of our knowledge.
> 4. **Novelty vs. Simplicity**. It seems there is a legitimate disagreement over the relative importance of novelty in methods versus simplicity and strong empirical results. We argue that the simplicity and effectiveness of contrastive learning is a key advantage over some complicated and heavy models, and applying it to capture class-wise knowledge is a novel and valuable contribution, supported by experimental results. We hope that in the future, the community will build on these blocks.
>
> **[Comment 2. Theoretical Insight]**
>
> Thanks for your valuable feedback. Your comments are very insightful and push us to further improve our paper. We will offer the theoretical insight during revision:
> - Minimizing the objective of class-wise attributors is maximizing a lower bound of the conditional mutual information between the saliency maps and target predictions, conditioning on the class labels.
> - Minimizing the objective of contrastive learning is maximizing a lower bound of the mutual information between the latent graph representations of two graphs within the same class.
> - Minimizing the objective of fine-tuning is maximizing a lower bound of the mutual information between the explanatory subgraph and target prediction of individual instances.

---

### Official Review · Reviewer_6Ffx · 2021-07-19

**Rating:** 6
**Confidence:** 4

**Summary:**

This paper presents a generative model to explain graph neural networks. The framework contains two steps: pre-training and fine-tuning. The first pre-training step trains edge probabilities in the adjacency matrix to capture class-wise saliency, followed by a contrastive learning by encouraging the similarity between two different adjacency matrices when they have the same class output. The second fine-tuning step employs  stochastic dropout to the edges based on the probabilities learned in the pre-training step while maximizing the mutual information between sampled subgraphs. Experiments on four datasets show consistent improvement over recent competing baselines, including SA, GNNExplainer, PGExplainer, and PGM-Explainer.

**Limitations And Societal Impact:**

I appreciate the authors discussing the limitation of the proposed method on the mapping between captured knowledge and the structure representation [13]. It would be great for the authors to further address the weaknesses mentioned above, especially on why the fine-tuning stage actually captures local patterns.

**Main Review:**

Strengths
* The task being addressed here is of high impact on how to explain learned graph neural networks, and can potentially help the community design better graph architecture in the future when we have deeper understanding of what the graph networks learn.
* Thorough literature survey and background knowledge. I especially like Sec 4.1 where the authors give a one-sentence summary of each recent competing method. This really helps the readers get up to speed and delve deep into the core experimental results.
* Strong performance in terms of usual metrics: ACC-AUC and recall. Figure 3 and 4 are also helpful in comprehending what the proposed loss functions capture qualitatively. Sec 4.2 is also well written in a sense that it gives a detailed walk through of the experimental results.

Weaknesses
* Despite the good metric numbers in tables, I still do not fully capture why the proposed two-stage pipeline can capture the claimed global patterns and local features. It seems like the only justification in the submission is in Figure 3 column 4, where the authors claim the explainer graph does not cross the MNIST digit and background. The authors did not perform the same pre-training / fine-tuning qualitative comparison in Figure 4 on Mutagenicity dataset. Based on the presented material, it is hard to fully believe that the proposed fine-tuning step can really capture the high local accuracy.
* Based on the formulation in Eq. 9, the proposed fine-tuning for local explainability is essentially a stochastic version of edge dropout based on the learned probability matrix in the pre-training step. Therefore, it is understandable that dropout would give a reasonable amount of performance boost, especially when the training data is relatively small. This is also reflected in Table 1 that most of the performance boost is contributed by the contrastive learning. Again, the overall observation does not tell the story that pre-training captures the global patterns while fine-tuning (edge dropout) captures the local patterns.
* Why does the fine-tuning (edge dropout) works well on MNIST but not that significant on other datasets?
* Figure 3 does not speicicy which column represents what digits. It is extremely hard for readers to understand it.
* Typo: line 55: show it effectiveness -> its, line 86: one extra period in the end of the sentence.


**Time Spent Reviewing:**

8

---

> ### Author Response · Authors · 2021-08-10
> **Response to Reviewer 6Ffx**
>
> Thank you for the valuable feedback. Your comments on the global & local explainability are very insightful and push us to further improve our clarity. Below please find our responses to your concerns.
>
> **[Comment 1. Reasons & Evidences of Capturing Multi-grained Explainability]**
>
> We clarify more about this from several perspectives.
>
> 1. We summarize the reasons and empirical evidence of capturing global and local explainability in the table below.
>
> |        |       Pre-training    |        Fine-tuning      |
> |--------|:---------------------|:-----------------------|
> |**Reasons** | For all graph instances within the same class, we assign them with a class-wise attributor, whose model parameters are shared. Moreover, we adopt contrastive learning to distill the class-wise knowledge by encouraging the agreement of graphs within the same class. As a result, the class-wise knowledge is captured by these class-wise attributors, referring to the global patterns. | For a single graph instance, we use its saliency map derived from the well-trained class-wise attributor and then further refine the saliency map towards its local fidelity (Eq. 8). As a result, the specific local patterns are refined as the explanatory subgraphs. |
> |**Evidences**| (1) Ablation results: Table 1; (2) Qualitative Results: Row 1 in Fig. 3.|  (1) Ablation results: Table 1 and Table 2; (2) Qualitative Results:  Fig.3 and Fig. 4. |
>
> 2. To clearly show the effectiveness of three key model components (class-wise attributors, contrastive learning, and fine-tuning), we extract the information from Table 1 in our paper and calculate their contribution scores to ReFine's ACC-AUC improvement over PG-Explainer. The contributions are summarized in the table below. Clearly, fine-tuning plays a significant role in improving the predictive accuracy and contributes on average 19.6% to the improvements, which is near the contribution score (22.4%) of contrastive learning. Hence, we can safely conclude that fine-tuning works well towards adaptively justifying the global patterns.
>
> | | Mutagenicity | VG-5    | MNIST   | BA-3motif |   Average |
> |---|-----|---------|---------|-----------|---------|
> | contribution of class-wise attributors | 79.3\%       | 81.5\%  | 16.7\%  | 54.5\%    | 58.0\%  |
> | contribution of contrastive learning  | 17.6\%       | 12.1\%  | 46.2\%  | 13.6\%    | 22.4\%  |
> |contribution of fine-tuning | 3.1\%        | 6.4\%   | 37.1\%  | 31.8\%    | 19.6\%  |
>
>
> **[Comment 2. Fine-tuning Behaves Differently across Datasets]**
>
> Thanks for bringing this to our attention. Fine-tuning works better on MNIST and BA-3motif, than on Mutagenicity and VG-5. Specifically, according to the table above, the contribution scores of fine-tuning to ReFine's improvements over PG-Explainer are 37.1%, 31.8%, 3.1%, and 6.4% on MNIST, BA-3motif, Mutagenicity, and VG-5, respectively. One possibility is the existence of rich node features in Mutagenicity and VG-5. With the assistance of node features, the global patterns might be well-captured during pre-training, thus leaving little space for the local patterns to improve. Overall, we argue that the effectiveness of fine-tuning is justified well, and we will thoroughly improve the clarity in our updated manuscript.
>
> **[Comment 3. Clarity of Figure 3]**
>
> We apologize that we did not make Figure 3 clear. The digits in the block are "0", "2", "8" from left to right. We will specify them during revision.
>
> **[Comment 4. Typos]**
>
> We sincerely appreciate your comments and will revise them in our updated manuscript.

---

> > ### Comment · Reviewer_6Ffx · 2021-08-27
> > **fine-tuning and local explainability**
> >
> > Thanks to the authors for preparing the answers. Again the proposed fine-tuning step for local explainability is essentially a stochastic version of edge dropout based on the learned probability matrix in the pre-training step. Besides the qualitative numbers, it is still hard to justify intuitively why edge dropout brings out the local explainability. I would like to keep my original rating.

---

### Official Review · Reviewer_TzAp · 2021-07-23

**Rating:** 6
**Confidence:** 4

**Summary:**

The authors improve the graph explainability by two methods. First is introducing the contrastive loss in the embedding space. The second is a random sampling to improve the original greedy top-k selection. The experiments are well done on different benchmarks with numerical results as well as qualitative visualization.

**Ethical Concerns:**

Yes

**Limitations And Societal Impact:**

Yes

**Main Review:**

* From Table 1, it seems the performance improvement are largely from the contrastive loss.  Recently, contrastive loss has been demonstrated to be effective in different applications, including graph learning, which makes the contribution relatively less novel. On the other hand, could you explain more how do you implement the class-wise attributor? Did you just simply train c attributors for each class? if so, is it a fair comparison that you use more parameters?

* Although we see some improvements from Refine-FT to ReFine in Table 1. However, if I understand correctly, PGM is very close to ReFine-CT, except for the finetuning. Comparing PGM and ReFine-CT, we don't see much advantage of having finetuning. Could you comment on this?

* The description from line 167 to line 181 could be improved. How do you sample "random edges"? How do you decide "\rho"? What do you mean by decrease exponentially with t?  How sensitive are those parameters? It's known that the the greedy selection is suboptimal in many problem by discarding the dependency. Having so many degree of freedom allows us to find some sweet spot to be better than greedy. Some ablation and robustness study is necessary to make the proposed finetuning steps convincing.

**Time Spent Reviewing:**

2

---

> ### Author Response · Authors · 2021-08-10
> **Response to Reviewer TzAp**
>
> Thank you for the comments. We offer the clarifications below to solve your concerns.
>
> **[Comment 1. Implementation of Class-wise Attributor]**
>
> Thanks for bringing the model size to our attention. We apologize that we did not make this clearer, and we conduct additional experiments. Here is the detailed responses and experimental results: (1) Indeed, we associate one attributor model to each class, which focuses on capturing the class-aware knowledge; (2) We argue that the superiority of our ReFine framework is attributed mainly to our novel pre-training & fine-tuning paradigm, rather than more model parameters. That is, the unsatisfactory performance of baselines, especially PG-Explainer, is ascribed to the limitation of the conventional training paradigm rather than the fewer model parameters. To justify this point, we conduct additional experiments and report the results in the table below, where PG-Explainer* is the PG-Explainer baseline with the same parameter size to our ReFine (i.e., $c$ times as the one reported in our paper). Clearly, our ReFine still outperforms PG-Explainer by a large margin across all datasets, under the fair comparison in terms of parameter scale. This verifies our argument that the superiority of ReFine comes mainly from our pre-training & fine-tuning paradigm, instead of more parameters.
>
> |                  | Mutagenicity | VG-5    | MNIST   | BA-3motif | BA-3motif |
> |------------------|--------------|---------|---------|-----------|-----------|
> |                  | ACC-AUC      | ACC-AUC | ACC-AUC | ACC-AUC   | Recall@5  |
> | PG-Explainer*[7] | 0.685        | 0.818   | 0.528   | 0.581     | 0.290     |
> | ReFine           | 0.955        | 0.914   | 0.636   | 0.630     | 0.304     |
>
> **[Comment 2. Novelty]**
>
> We respectfully disagree with your statement on "the novelty of using contrastive learning". Below, we clarify more about our novelty from several perspectives:
>
> 1. **Task Novelty**. Distinct from previous efforts either on global explainability or local explainability, our work focuses on multi-grained (multi-level) explainability that is less explored in the literature.
> 2. **Paradigm Novelty**. Towards multi-grained explainability, we introduce the pre-training & fine-tuning paradigm in the explanation generation, which seamlessly unifies the global and local explainability. Therefore, our main technical contribution lies in the novelty of tasks and paradigms.
> 3. **Novelty in Explainability**. There exist a handful of solutions towards class-aware knowledge. In our pre-training phase, we adopt contrastive learning, the simple but powerful tool, to achieve this goal. More importantly, although contrastive learning is used in graph learning, it has never been used in the literature of explainability, to the best of our knowledge.
> 4. **Novelty vs. Simplicity**. It seems there is a legitimate disagreement over the relative importance of novelty in methods versus simplicity and strong empirical results. We argue that the simplicity and effectiveness of contrastive learning is a key advantage over some complicated and heavy models, and applying it to capture class-wise knowledge is a novel and valuable contribution, supported by experimental results. We hope that in the future, the community will build on these blocks.
>
> **[Comment 3. Comparison of PG-Explainer and ReFine-CT]**
>
> We apologize that we did not make the introduction of ReFine-CT and ReFine-FT clear. We will provide a more straightforward explanation of these variants and highlight their differences from PG-Explainer during revision. Here is the detailed clarification: ReFine-CT simultaneously removes the contrastive learning from the pre-training phase and disables the fine-tuning phase. To improve the clarity, we summarize the distinctions of ReFine's variants (ReFine-CT, ReFine-FT) and PG-Explainer in the first table below.
>
>
> | | class-wise attributors | contrastive learning | fine-tuning |
> |---|------|-----|----------|
> | PG-Explainer | - | - | - |
> | Refine-CT | $\surd$ | - | - |
> | Refine-FT |$\surd$|$\surd$|-|
> | Refine |$\surd$|$\surd$|$\surd$|
>
>
> To obtain the importance of each model component to ReFine's performance, we extract the information from Table 1 in our paper and calculate the contribution scores as shown in the table below.
>
> | | Mutagenicity | VG-5    | MNIST   | BA-3motif |   Average |
> |---|-----|---------|---------|-----------|---------|
> | contribution of class-wise attributors | 79.3\%       | 81.5\%  | 16.7\%  | 54.5\%    | 58.0\%  |
> | contribution of contrastive learning  | 17.6\%       | 12.1\%  | 46.2\%  | 13.6\%    | 22.4\%  |
> |contribution of fine-tuning | 3.1\%        | 6.4\%   | 37.1\%  | 31.8\%    | 19.6\%  |
>
> The table presents empirical supports for the significance of fine-tuning, which contributes 37.1%, and 31.8% to ReFine's improvements over PG-Explainer on MNIST and BA-3motif datasets, respectively. One possible reason that fine-tuning contributes only 3.1% and 6.4% in Mutagenicity and VG-5 is the existence of rich node features in these two datasets. With the assistance of node features, the global patterns might be well-captured during pre-training, thus leaving little space for the local patterns to improve. Overall, we argue that the effectiveness of fine-tuning is justified well, and we will thoroughly enhance the clarity in our updated manuscript.
>
>
> **[Comment 4. Description of Fine-tuning]**
>
> In Equation (9), we formulate the the selection module as $\mathbf{S}^{(c)}=\mathcal{H}(\mathcal{G}_{att}^{(c)}, f, c, \rho, t)$, where $\rho$ is the user-defined edge number. Specifically, we set $\rho$ equal to 10%, ...,100% of the total number of edges, respectively, when computing ACC-AUC.
>
> The sampling function $\mathcal{H}$ samples edges from a probability distribution, which is normalized from the saliency map of the attentive graph $\mathcal{G}_{att}^{(c)}$. We follow the simulated annealing algorithm, where the probability for $\mathcal{H}$ to accept a "worse" (not within the first k optimals) edge is proportional to $e^{-at}$, where $a$ is a hyper-parameter. We will update the sensitivity of the parameter during revision.
>
> **Reference:**
>
> 1. Dongsheng Luo, Wei Cheng, Dongkuan Xu, Wenchao Yu, Bo Zong, Haifeng Chen, and Xiang Zhang. Parameterized explainer for graph neural network. In NeurIPS 2020.
> 2. Minh N. Vu and My T. Thai.  Pgm-explainer: Probabilistic graphical model explanations for graph neural networks. In NeurIPS 2020.
> 3. Jeroen Kazius, Ross McGuire, and Roberta Bursi. Derivation and validation of toxicophores for mutagenicity prediction.Journal of medicinal chemistry, 48(1):312–320, 2005.
> 4. Kaspar  Riesen  and  Horst  Bunke.   Iam  graph  database  repository  for  graph  based  pattern recognition and machine learning. In SPR and SSPR, pages 287–297, 2008.

---

> > ### Comment · Reviewer_TzAp · 2021-08-23
> > **Raised score**
> >
> > Thanks for the clear explanation and the addition experiments. I increased my score from 5 to 6. Please ensure to include those discussion in the revision.

---

### Decision · Program_Chairs · 2021-09-28

**Decision:**

Accept (Poster)

**Comment:**

The reviews for this paper were overall positive, commenting on the high impact, strong performance,  good motivations, and easy to implement method. After the authors helped address some of the reviewers’ concerns (particularly around novelty), all reviewers are recommending acceptance.

We recommend to take the reviewers' comments and suggestions into account while preparing the camera ready final version of the paper.  Particularly any intuition on why edge dropout helps local explainability would be beneficial.

**Consistency Experiment:**

NeurIPS has a long history of experimentation. In 2014, NeurIPS ran an experiment in which 10% of submissions were reviewed by two independent committees to quantify the randomness in the review process. This year, we repeated a variant of this experiment to see how the quality of the review process has changed over time.  This paper was part of the experiment and was therefore assigned to two committees (consisting of reviewers, an Area Chair, and a Senior Area Chair) that reached independent decisions.  If both committees made the same recommendation, this recommendation was followed. If a single committee recommended acceptance, the paper was accepted (with the exception of a few cases in which the other committee identified what we considered a fatal flaw, e.g., an error in a key result).

This copy’s committee reached the following decision: **Accept (Poster)**

The other committee assigned to the paper recommended **Reject**.  You can find the other set of reviews, along with any follow up discussion with the authors here:
https://openreview.net/forum?id=e5vrkfc5aau